# The Effect of Food Vouchers and an Educational Intervention on Promoting Healthy Eating in Vulnerable Families: A Pilot Study

**DOI:** 10.3390/nu14234980

**Published:** 2022-11-23

**Authors:** María L. Miguel-Berges, Andrea Jimeno-Martínez, Alicia Larruy-García, Luis A. Moreno, Gerardo Rodríguez, Isabel Iguacel

**Affiliations:** 1Growth, Exercise, Nutrition and Development (GENUD) Research Group, University of Zaragoza, C/Pedro Cerbuna 12, 50009 Zaragoza, Spain; 2Instituto Agroalimentario de Aragón, 50013 Zaragoza, Spain; 3Instituto de Investigación Sanitaria Aragón (IIS Aragón), 50009 Zaragoza, Spain; 4Centro de Investigación Biomédica en Red de Fisiopatología de la Obesidad y Nutrición (CIBEROBN), 50009 Zaragoza, Spain

**Keywords:** vulnerable groups, nutrition education intervention, food vouchers, obesity, health, healthy eating

## Abstract

Cost has been reported as the main barrier to healthy eating in vulnerable groups. We aimed to evaluate the effect of a nutrition education intervention on adherence to Mediterranean Diet and health when providing food vouchers. This pilot study has a randomized control trial design. We included 66 vulnerable users from the Red Cross of Zaragoza (Spain). Intervention and control group individuals received 120 euros/month of food vouchers over 3 months to be spent in supermarkets (60 euros/month if under 12 y) plus a 10-week nutrition education program for the intervention group. Family food purchases were assessed using electronically recorded supermarket-obtained transactions. During and at the end of the intervention the percentage of healthy food was higher in the intervention than in the control group. Once the nutrition education was over, differences between groups dissipated. In the intervention group, health parameters improved, particularly weight-status, lipids, and liver enzymes. Control participants gained weight, although lipid and liver enzymes improved. Blood pressure and HbA1c did not improve in either the intervention or the control group. In conclusion, providing unrestricted food vouchers to vulnerable groups to increase healthy food consumption appears to be insufficient and should be accompanied by medium-long term nutrition education.

## 1. Introduction

The consumption of a diet with a high intake of energy-dense and low nutrient-dense foods is a major risk factor for weight gain, obesity, and associated metabolic and cardiovascular diseases morbidity and mortality [1]. Families with greater socioeconomic disadvantages, such as lower income, lower educational level, or living in socioeconomically disadvantaged neighborhoods, are more likely to have overweight and poorer health outcomes compared to families with less disadvantage [2,3]. As such, the price and affordability of food are key determinants in the choice of products included in the shopping basket, with people of lower socioeconomic status being the most sensitive to food prices [4].

In 2021, The Food and Agriculture Organization of the United Nations (FAO) published a report conveying that the cost of food and beverages and their affordability are key determinants of malnutrition, including undernutrition and obesity, globally [5]. While some studies have suggested that a healthy diet is not more expensive than less healthy options [6,7], this report estimated that the price of a healthy diet (rich in whole grains, fruits, vegetables, legumes, nuts and seeds, and dairy products) costs 60% more than diets that only met essential nutrient requirements based on a limited number of foods. It also estimated that 3 billion people could not afford the cost of a healthy diet, without taking into account recent global events such as the SARS-CoV-2 pandemic that severely impacted employment and household incomes [8,9].

The perceived high price of healthy foods compared to unhealthy foods is a major barrier to eating a balanced diet, particularly for vulnerable groups [10]. Moreover, unhealthy foods seem to be more likely to be chosen by individuals because supermarkets often encourage the purchase of unhealthy foods in their catalogs, at checkout counters, or in shelf space [11]. Temporary price promotions and discounts on unhealthy foods contribute to an increase in the amount of food purchased and consumed in the short term, especially unhealthy foods containing excessive amounts of sugar, saturated fat, salt, and/or alcohol [8,12,13].

Low socio-economic status (SES) has been associated with less healthy dietary habits [14]. Although the reasons remain unclear, some of the motivations pointed out by vulnerable groups to having unhealthy dietary habits are their higher priority for price and familiarity, and their lower priority for health [15].

Food vouchers have been used as a tool to reduce health inequalities and improve barriers to a healthy diet. However, the results on the effects of these food subsidy programs on vulnerable populations are controversial. Indeed, while some studies suggest that the mere provision of food vouchers is sufficient to improve dietary habits, others suggest that the possible positive effect is mediated by an educational intervention being conducted simultaneously [16,17,18].

In addition, many of the previous studies have included food vouchers targeted only to the purchase of certain products (e.g., only fruit and vegetables) with less financial support. The lack of an educational intervention together with financial aid could also be one of the possible reasons for the inconclusive results in favor of these social interventions [17,19].

Moreover, most of the studies including an intervention in vulnerable groups have evaluated the effect that either education or incentives can have on their dietary patterns, assessed subjectively with food frequency questionnaires (FFQ) or 24 h-dietary recalls (24-HDR), instead of using more accurate tools [17]. Thus, these dietary assessment methods are generally limited by self-report biases [20]. The present paper uses electronically recorded, supermarket-obtained transactions as an objective measure of family food purchases.

Therefore, the aim of this study was to assess if a 10-week intervention, including both food vouchers and nutritional and health education, improves diet quality, and, consequently, the health of vulnerable participants in Zaragoza (Spain). Additionally, both groups (intervention and control) received, 8-weeks after the above 10-week intervention, only food vouchers (no educational intervention) to assess whether the possible positive effect of this intervention was maintained over time.

## 2. Materials and Methods

### 2.1. Study Design

This pilot study has a randomized pretest–post-test experimental design. The project was designed to improve the dietary habits (increasing the intake of fruits, vegetables, legumes, and nuts) and, consequently, the health of vulnerable groups. The intervention was implemented in Zaragoza, a north-east city in Spain, that is the capital of the Zaragoza Province and of the autonomous community of Aragon [21]. According to the latest data published [22], Zaragoza is the fifth largest city in Spain, with 714.058 inhabitants, 52% of whom are women, and 16% are immigrants.

### 2.2. Study Population

Adults over 18 years old belonging to the program of extreme vulnerability of the Red Cross of Zaragoza were invited to participate in this pilot study. This denomination is used by the Red Cross to refer to people who, due to their socioeconomic level, live in a situation of extreme poverty [23].

After a general meeting, in which members of the Red Cross explained the project to a total of 24 adults, 21 adults (and their family members) agreed to participate. The sample ultimately included 66 children and adults. The entire recruitment process was performed through the Red Cross.

Participants were randomly selected using the free OxMaR software for randomization of studies. Only one person of the household unit (usually the mother or the woman if they were a couple) was randomized and included in either the control group or the intervention group. Then all members belonging to that household were automatically included in the control group or the intervention group. Thus, the whole household unit was always in the same group (either control or intervention). Finally, 32 individuals were assigned to the control group and 34 individuals to the intervention group.

The inclusion criteria established were children and adults between 3 and 80 years of age with at least one adult per family unit that could speak and understand Spanish, French, or English. Adults with severe illness or cognitive impairment were excluded from the study.

### 2.3. Intervention (Independent Variable)

Control group: the control group received food vouchers for 14 weeks (from mid-October to December 2021 and additionally for one more month in March 2022). Adults and children 12 years of age or over received 120 euros per month in food vouchers, whilst children between 3 and 11 years of age received 60 euros per month. As an example, a family consisting of two adults, one child aged 12 years, and another child aged 8 years received a total of 420 euros per month, making a total of 1470 euros during the whole project.

Intervention group: Like the control group, the intervention group received the same quantity of money in food vouchers (from mid-October to December 2021) plus one more month in March 2022 when the education program was over. In addition, the intervention group received a healthy lifestyle education program for 10-weeks (from mid-October to December 2021). The educational intervention covered a wide range of relevant topics, including the importance of healthy eating in disease prevention, and the association of processed food and cardiovascular diseases, the food pyramid, Mediterranean Diet, planning healthy snacks and dinners, food handling and hygiene, and the importance and planning of physical activity. These sessions were conducted two times per week by two qualified dietitians and a physician during the 10-week intervention and had an average duration of 50 min each. The intervention group also received examples of personalized simple menus based on dietary recommendations, and fitting with the indicated budget. Different options of healthy, easy-to-cook, quick, and affordable daily recipes, fully customized to their cardio-metabolic condition and cultural diets and maximizing their economic budget, were discussed with the individuals or family units. In addition, they received health advice personalized to their physical and mental status with a direct communication via WhatsApp with a physician during the intervention time.

To sum up, both groups (control and intervention) received the same amount of money in food vouchers. In contrast to the control group, the intervention group received health and nutrition education during a 10-week intervention from mid-October to December 2021. Additionally, in March 2022, both groups received only food vouchers for one month to check whether the effect of the educational intervention was maintained in the intervention group (Figure 1).

Since there were no recent studies conducted in Spain that had calculated the specific cost of a healthy diet, a group of nutritionists and physicians developed several menus providing the required macro- and micronutrients, but minimizing the costs and using prices as of October 2020 of an average supermarket in Spain. The average food cost per month per person (over 11 years old) in Spain to eat healthily was estimated at 120 euros using 50 healthy food items with online prices collected from one retailer website (https://www.dia.es/compra-online/, accessed on 21 July 2021) during summer 2021. For children under 11 years old, this budget was set at 60 euros per month.

For that reason, all participants (from both the intervention and the control group) received unrestrictive voucher cards to obtain food or drinks at any supermarket of the DIA group in Zaragoza to finance family meals for 14 weeks (from mid-October to December 2021 and March 2022).

### 2.4. Outcome Measures

#### 2.4.1. Main Outcome: Adherence to the Mediterranean Diet

To determine the degree of adherence to the Mediterranean Diet in adults, a specific short questionnaire validated for the Spanish population and used by the Prevention with Mediterranean Diet (PREDIMED) group was used [24]. A trained dietitian administered this 14-item questionnaire. Briefly, for each item, a score of 1 or 0 was assigned. Once the values for the 14 items were obtained, they were added up. Thus, the score ranged from 0 to 14, in which 0 points meant null adherence and the 14 points meant the highest adherence in adults.

In children, adherence to the Mediterranean Diet was assessed by a questionnaire of 16 items adapted to children from that previously described and used in the PREDIMED study [24]. Compliance for each of the 16 items was scored with 1 point. Thus, the score ranged from 0 to 16, in which 0 points meant null adherence and the 16 points meant the highest adherence in children.

#### 2.4.2. Dietary Assessment

Food and drink consumption for all individuals was registered from all purchases made in participating supermarkets from October 2021 to March 2022. However, to capture possible purchases made outside the participating supermarkets, participants were additionally asked to report purchases made in other supermarkets.

#### 2.4.3. Secondary Outcomes

Anthropometric and blood pressure measurements, blood tests, questionnaires, and the health and nutrition education program were carried out at the facilities of the Red Cross of Zaragoza. During the first two weeks of October 2021, we collected fasting blood samples of participants. Moreover, each subject’s weight, height, waist and hip circumference, body composition (fat mass, lean mass, and total body water), and blood pressure were measured. Also, participants were interviewed to complete different questionnaires to collect sociodemographic information, dietary intake, and information about their physical activity and sedentary behaviors. When children were under 12 years of age, the mother or the father responded the questionnaire. All these data were collected again at the end of December 2021.

##### Anthropometric Measures and Body Composition

The anthropometric measures and body composition variables studied were body weight (kilos) and height (meters), BMI (kg/m^2^), BMI z-score, fat (kilos), fat mass index, water (percentage), lean mass (kilos), waist, waist/height, and hip (cm).

Body weight and height were measured to the nearest 0.01 kg and 0.01 mm, respectively. Body Mass Index (BMI) was also calculated as body weight in kilograms divided by body height in meters squared (kg/m^2^) and categorized into underweight (if BMI < 18.5), normal weight (18.5–24.9), overweight (25–29.9), and obesity (30 or greater) [25]. BMI for children and adolescents was converted to an age- and sex-specific z-score using the extended IOTF criteria [26]. To assess weight and body composition, the Tanita MC780SMA portable beam scale was used. During measurement, the subjects remained standing in the center of the scale without support and with their weight equally distributed on both feet covering the metal plates of the Tanita base. To take the measurement, the participant stood without socks and shoes and with as little clothing as possible on the platform and held the cuffs with both hands.

Fat mass, lean mass, and percentage of body water were estimated using the information obtained from the Tanita scale. Fat mass index (FMI) was calculated [kg body fat derived from the percentage of body fat from the Tanita/height (m^2^)].

The SECA 213 portable measuring rod (accuracy 1 mm SECA, Hamburg, Germany) was used to determine height. The subject had to stand with heels together and buttocks and upper back in contact with the scale. The head was to be placed in the Frankfort position. The Frankfort plane is obtained when the Orbitale^®^ point (lower edge of the eye socket) is in the same horizontal plane as the Tragion^®^ point (the upper notch of the tragus of the ear). A deep inspiration was performed, and the measurement was taken.

Waist and hip circumferences were measured in triplicate using an anthropometric tape (SECA 200) [27].

Regarding the weight objective, it was considered as achieved when participants with overweight and obesity lost weight after the intervention, those underweight gained weight and those of normal weight lost or maintained weight.

##### Socio-Demographic Variables

Sex and age were collected. Date of birth was reported by one of the parents/caregivers of children or by adolescents and adults by themselves. Age was computed based on date of birth and the date when the questionnaire was completed. According to the World Health Organization (WHO), age was categorized into child (3–9 y), adolescent (10–19), adult (20 or older) [28]. Migrant status was considered if participants were born in a different country than Spain. Children born in Spain but with both parents born in a country other than Spain were also assigned as migrants. Education of the parents/caregivers and partners was self-reported in the primary caregiver’s questionnaire. The highest level of educational attainment of the mother and the father was dichotomized into low and high (more than 14 years of education) SES, which distinguishes families with an adult who has completed medium or higher education, such as college or university training from other families [29]. Adults reported their occupational status in the last six months assessed by six response options: homemaker, work full-time, work part-time, unemployed, full-time student, and retired or pensioner. Occupation classifications were combined to ‘employed’ (work full time or part time) vs. ‘unemployed’ (homemaker unemployed, full-time student and retired or pensioner). The employed category was used if at least one of the adults in the family unit was employed. Regarding the family structure, a traditional family was considered when children or adolescents were living with both biological parents. Other types of families were considered as non-traditional.

##### Blood Pressure

Systolic blood pressure (SBP) and diastolic blood pressure (DBP) in mmHg were measured with an automatic oscillometric device (OMRON M6). All participants were asked to sit for at least 5 min before the measurement was taken. Two measurements were taken with a 2-min interval, plus a further measurement in case of a >5% difference in blood pressure between the first two readings. The average of the two (or three) measurements was used for statistical analysis.

##### Blood Analyses

Blood samples were taken by a hematologist after 8 h of fasting. Blood samples were obtained at 9 am at pre- and post-intervention. Further, all participants were required to fast for 8 h before extracting their blood sample. Approximately 10 mL of blood was extracted from the median antecubital vein and stored in an EDTA tube. The plasma was centrifuged for 15 min at 3000 rpm at 4 °C. Thereafter, the samples were stored frozen for subsequent analysis. All samples were measured directly by an automatic hematology analyzer. The laboratory carries out quality control according to standard procedures.

We collected a complete blood count test measuring white blood cell count, white blood cell types, red blood cell count, hematocrit, hemoglobin, red blood cell indices, platelet count; other parameters, such as erythrocyte sedimentation rate (after one hour and two hours and Katz index), serum iron metabolism, and a comprehensive metabolic panel, including uric acid, blood urea nitrogen (BUN), creatinine, bilirubin, liver enzymes, including ALP (alkaline phosphatase), Glutamate-Pyruvate Transaminase (GPT), glutamic oxaloacetic transaminase (GOT), and Gamma-glutamyl transferase (GGT); lipid levels (including triglycerides (TG), total cholesterol (TC), high-density lipoprotein cholesterol (HDL-C), and low-density lipoprotein cholesterol (LDL-C), fasting glucose, and glycated hemoglobin A1c (HbA1c).

### 2.5. Statistical Analysis

Continuous variables were presented as mean ± standard deviation (SD) and categorical variables as percentage. For these variables, comparisons between groups were performed using the independent Student’s t-test for 2 groups. Before applying the post-hoc test, we tested the homogeneity of the variances between the groups with Levene’s test. For descriptive analysis, the chi-square test of independence was used to examine possible differences in the study population according to whether they belonged to the control group or the intervention group. The significance level was set at *p* < 0.05.

Moreover, percentage change was calculated as the change in value divided by the absolute value of the original value, multiplied by 100, that is, the difference between parameters at the end of December (after the 10 weeks-intervention) and before the intervention (in mid-October).

To check how much difference there was between the averages of the intervention group and control group, the mean difference was calculated as the absolute difference between the mean value in both groups with the independent samples *t*-test for equality of mean. The significance level was also set at *p* < 0.05.

All data analyses will be carried out using the SPSS version 26.0 statistical package.

### 2.6. Ethical Considerations

The study was developed in compliance with the ethical principles of the 1964 Declaration of Helsinki, revised in 2000 in Edinburgh. The standards of good clinical practice of the International Conference on Harmonization for Good Clinical Practice were respected. All participants who wished to participate in the study signed an informed consent form on acceptance of participation; in the case of minors, informed consent was obtained and signed by parents or legal guardians.

This project was approved by the Research Ethics Committee of the Autonomous Community of Aragón (CEICA), C.P.-C.I. PI20/541 Acta No 17/2021.

Clinical Trial Registration—URL: http://www.clinicaltrials.gov (accessed on 9 September 2022). Unique identifier: NCT05539222.

## 3. Results

Table 1 shows the main sociodemographic characteristics of the sample. A total of 66 individuals participated, of which 32 belonged to the control group and 34 to the intervention group. Overall, 59.1% of the subjects were women. Of the participants, 81.8% had a migrant origin (with either Maghrebian or Latin-American origin). Most participants had a low educational level (83.3%), an unemployed status (69.7%), and a traditional family structure (63.6%). Regarding weight status, 56.1% of the participants were overweight or obese (with a higher percentage in the intervention group). When differentiating between adults and children or adolescents, the percentage of overweight was 35.3% and obesity 41.2% in adults, and in children and adolescents, 15.6% and 18.8%, respectively. In relation to whether the target weight was achieved, 75% of the participants in the intervention group achieved their target weight (that is, those who were overweight or obese lost weight after the intervention, those who were underweight gained weight, and those of normal weight lost or maintained their weight).

To assess the effect of the intervention, the main biochemical and anthropometric parameters and blood pressure measured pre- and post-intervention were compared; these data are shown in Table 2, Table 3 and Appendix A. After the educational program, the mean weight of the subjects in the intervention group decreased, while those in the control group gained weight. The percentage of fat mass increased and lean body mass improved, respectively, in both groups, however, the percentage of body water remained almost stable. The average waist and hip circumference decreased in subjects in the intervention group but increased in those in the control group.

The liver function markers improved in both groups (control and intervention), except for the GOT marker, which worsened in the control group.

Lipid levels (particularly HDL-C and LDL-C) improved in both groups. TG and TC also decreased in the intervention group, but this effect was not seen in the control group.

HbA1c increased in both groups, but a higher percentage was found in the intervention group.

Finally, DBP increased in both groups and SBP increased in the intervention group and decreased in the control group.

After 10 weeks of an educational program, adults who were randomly assigned to the intervention group had statistically significantly greater adherence to the Mediterranean Diet (1.50 vs. 3.88, mean change difference, 2.38), greater weight loss (0.91 vs. −1.58, mean change difference, −2. 49), greater BMI loss (0.36 vs. −0.59, mean change difference, −0.95), greater fat loss (−0.70 vs. −2.21, mean change difference, −53.40), greater urea loss (2.93 vs. −3.35, mean change difference, −6.27), and greater triglyceride loss (27.10 vs. −26.30, mean change difference, −6.27), compared to the control group. Similarly, children and adolescents randomized to the intervention group had statistically significantly greater weight loss (1.28 vs. −0.33, mean change difference, −1.61), and increased DBP (0.18 vs. 7.97, mean change difference, 7.79) compared to the control group (Table 4).

In Table 5 the distribution of the frequency of purchase of the different food groups by family unit is shown.

A higher percentage of healthy food items such as fruits and vegetables, legumes, fish, nuts and seeds, and olive oil was consumed in the intervention group compared to the control group. On the other hand, more unhealthy items such as sugar, bakery products, cakes and confectionery, ready-to-eat dishes, sweetened beverages, or salty snacks were consumed in a lower percentage by the intervention group compared to the control group. Once the intervention was over, the intervention group still consumed a higher percentage of healthy food items and lower percentage of unhealthy foods compared to the control group. However, these differences between the two groups decreased after the end of the intervention.

## 4. Discussion

The aim of this study was to evaluate the effect of a 10-week health and nutrition education intervention in extremely vulnerable groups on adherence to the Mediterranean Diet and health when providing food vouchers. For this purpose, electronically recorded transactions obtained in the participating supermarkets were used as an objective measure of family food purchases. In addition, two months after the intervention was over, both groups (intervention and control) received only food vouchers (without an educational intervention) to assess whether the possible positive effect of the intervention was sustained over time.

Before the intervention, adherence to the Mediterranean Diet was statistically significantly higher in the control group than in the intervention group. However, after the intervention, both groups had a higher adherence to the Mediterranean Diet, even though the effect was only statistically significant for the intervention group. These results indicate that the effect of food vouchers alone seems to be beneficial, although the educational intervention seems to be the most important element in achieving greater adherence to the Mediterranean Diet in vulnerable families.

People from low SES have less healthy dietary habits, partly because of their higher priority for price and familiarity, and their lower priority for health as a motive for food purchases [15]. Based on prices of 2021 in Spain, we estimated that to eat healthfully, adolescents and adults needed a minimum of 120 euros per month. Thus, all participants from the intervention and control group received that quantity in food vouchers. This financial incentive allowed us to eliminate one of the main barriers that vulnerable groups have suggested to eating healthfully [10]. Families of low SES, and especially immigrant families [30], report difficulties in acquiring healthy foods, mainly fruits, vegetables, and dairy products [31], because of their high cost compared to ultra-processed products [32].

In Spain in 1987, 11% of the calories ingested in the diet came from ultra-processed products, while in 2007, this type of product accounted for 32% of dietary intake, practically tripling the percentage [33]. Today, young people consume only 1.2 serves of fruit and vegetables per day, far below the recommended five serves [33].

In our study, no restrictions regarding the food and drinks that could be chosen by participants were implemented. However, participants were recommended to buy healthy food, and to avoid alcohol and sugar sweetened beverages (SSB).

Our results showed that the food vouchers alone might not be sufficient to improve the diet or health parameters analyzed for the extremely vulnerable groups. These results are in agreement with a previous study carried out in a low-income population in France. This study conducted a 3-month program assessing the effects of food vouchers (only exchangeable for fresh fruit and vegetables) on the intake of fruits and vegetables and plasma levels of biomarkers. The group receiving both the food vouchers and the nutritional advice did not show either a higher increase in mean intake or a change in relevant biomarker levels compared with the group receiving only dietary advice. The results of this French study suggest the importance of dietary advice over food vouchers [17].

Nonetheless, it should be kept in mind that other studies have pointed out that providing nutrition education only or food vouchers only has limited effects on the health or feeding practices of the participants. Indeed, in a recent study conducted in Ethiopia, when provided both education and food vouchers together, child-feeding practices improved and stunting prevalence decreased [34].

In our study the intervention group had better dietary and health outcomes than the control group. After 10 weeks of an educational program, the adults who were randomly assigned to the intervention group had statistically significantly greater adherence to the Mediterranean Diet, greater weight loss, greater BMI loss, greater fat loss, greater urea loss, and greater triglyceride loss compared to the control group. Similarly, children and adolescents randomized to the intervention group had statistically significantly greater weight loss and BMI z-score, and increased DBP compared to the control group. These results suggest that the effect of an educational intervention seem to be higher in adults than in children and adolescents. Additionally, during the 10 weeks of the educational intervention, more than half of the products purchased corresponded to unprocessed or minimally processed products, such as fruits and vegetables, legumes, and fresh meat or fish in the intervention group.

After the 10 weeks’ intervention, and once the educational intervention was over, during the month of March 2022, both groups (intervention and control) received only food vouchers. The percentage of unprocessed or minimally processed food items purchased decreased in both groups, but specially in the intervention group. Also, in the intervention group, the percentage of unhealthy items, such as sugar sweetened beverages, sugar, bakery products, cakes and confectionery, or salty snacks, increased significantly compared to the control group.

Almost all health parameters improved for the intervention group. Specifically, 75% of the participants in the intervention group achieved the target weight according to their BMI, losing on average one kilo over the 10 weeks. Lean mass, waist circumference, and hip circumference also decreased.

In both groups (intervention and control), the lipid profile and liver function also improved. Contrary to expectations, blood pressure and HbA1c worsened, although mean values were all in range and there were no statistically significant differences when the effect of the intervention was analyzed.

One issue that differentiated the control group from the intervention group was that, unlike the latter, the former gained weight. In fact, most participants in the control group did not achieve their objective regarding their weight. These results are in line with previous investigations conducted in the United States [35,36]. These studies found that the Food Stamp Program (currently known as Supplemental Nutrition Assistance Program (SNAP)), which is a federal program that provides food-purchasing assistance for low- and no-income people, contributed to participants’ weight gain and higher BMI, especially female participants [35,36]. Additionally, the authors of this study found that the longer they participated, the greater their BMI [36]. Although the mechanisms between these programs and weight gain remained unclear, in order to prevent obesity in beneficiaries, some modifications have been suggested, such as creating vouchers or coupons only for buying fruit and vegetables, requiring vendors to offer only healthier options, or restricting the purchase of soft drinks and other unhealthy foods.

According to the studies carried out, SNAP participants generally have enough calories to sustain themselves and their calorie intake does not differ systematically from that of income participants or higher-income non-participants. However, some studies, including a systematic review, showed that SNAP participants had similarly low or significantly lower dietary quality than comparison groups [37,38]. Nonetheless, one study has suggested that patients receiving a food voucher, especially those who were born outside of the United States or who were limited English proficiency, purchased more fruits and vegetables and less SSB than national averages [39].

In the present study, even though the average age of the participants was under 30, 56.1% of the participants had overweight or obesity (with a higher percentage in the intervention group). Indeed, when differentiating between adults and children or adolescents, the percentage of overweight and obesity was 76.5% in adults and 34.1% in children and adolescents. Similar proportions of childhood overweight and obesity have been found in Spain [40], but a much higher percentage of overweight and obesity was found in the present adult sample of extremely vulnerable people compared to Spanish data [41].

The results of this pilot project show that economic support (i.e., food vouchers) to improve dietary habits and health of participants must be accompanied by an educational intervention, particularly in extremely vulnerable groups. Although after the 10-weeks education program the intervention group continued to buy more healthy products than the control group, the differences observed during the intervention period dissipated. In fact, when the educational intervention ended, and food vouchers were given again to both groups two months later, the intervention group’s diet worsened (decreasing the percentage of fresh products such as fruits and vegetables), while the control group’s purchasing conduct remained similar. These results can be explained by the fact that, although participants knew that all purchases being made could be identified, they may no longer have felt that they were being monitored. Thus, when the educational intervention ended, participants in the intervention group may have chosen the food they really wanted to buy. Another possible explanation could be the fact that the 10-week educational intervention was not long enough to establish healthy dietary behaviors, hence the need for a long-term educational intervention in these extremely vulnerable groups.

### Strengths and Limitations

To the best of our knowledge, this is the first study conducted in Europe in which a group of extremely vulnerable people received enough money to have a balanced diet for more than three months, including an educational health program for the intervention group. Moreover, we were able to record, electronically, supermarket-obtained transactions as an objective measurement of food purchases. Additionally, biochemical parameters, blood pressure, as well as anthropometric measurements were assessed.

Nevertheless, this study has also some limitations. Even though family units were randomly assigned to a control or an intervention group, participants belonging to the intervention group showed a higher vulnerability pattern compared to the control group (i.e., they had a lower educational status and higher levels of obesity), which could have had an impact on our results. Secondly, this pilot study included 66 participants from an extremely vulnerable group in the city of Zaragoza, Spain, therefore limiting the generalizability of the results. Thirdly, family food purchases, rather than individual food purchases were analyzed, which made changes at the individual level more difficult to observe.

## 5. Conclusions

This study aimed to assess the effects of a 10-week health and nutrition education intervention on adherence to Mediterranean Diet and health when providing food vouchers in extremely vulnerable groups. The percentage of healthy food items electronically registered was higher in the intervention than in the control group. Nonetheless, once the nutrition education was over, the differences between the intervention and control group dissipated. Regarding health parameters (including some biochemical parameters, blood pressure and anthropometric measurements), after a 10-week educational intervention, generally they seemed to improve in the intervention group, particularly in weight status, lipid profile, and liver function. On the other hand, the control group gained weight, although their lipid profile and liver function also improved. Blood pressure parameters and HbA1c did not improve in either the intervention or the control group.

In particular, our results suggest that public actions aimed at improving dietary patterns, achieving greater adherence to the Mediterranean Diet, or improving the health of participants should include active educational interventions (not merely passive ones such as handing out information leaflets), probably for six months or more, including follow-up and with the provision of food vouchers preferably restricted to the purchase of certain products (e.g., fruit, vegetables, pulses, nuts, dairy products, olive oil, etc.).

Future lines of research should include longer-term educational interventions and follow-up to obtain more accurate results for health policymakers, using objective dietary assessment tools. It is also expected that these studies will include greater tailoring and adaptation to the needs of these vulnerable groups.

In conclusion, our findings suggest that providing unrestricted food vouchers to vulnerable groups to improve health and increase the quantity of healthy food, such as fruits, vegetables, and nuts, appears to be insufficient and should be accompanied by medium-long term nutritional or health education.

## Figures and Tables

**Figure 1 nutrients-14-04980-f001:**
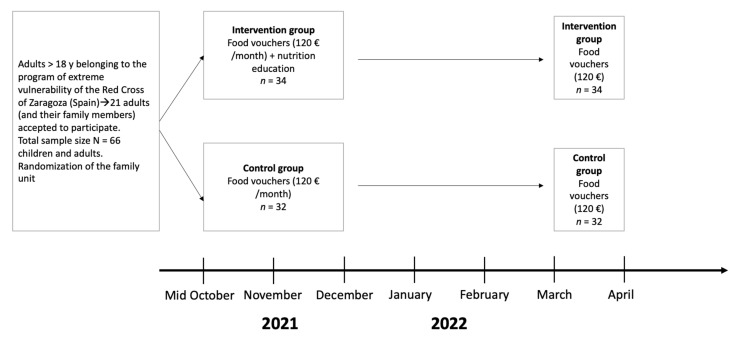
Study design.

**Table 1 nutrients-14-04980-t001:** Sociodemographic and anthropometric characteristics of the sample (children, adolescents, and adults).

Categorical Variables	N = 66 (%)	Group (%)		*p*-Value
		Control (*n* = 32)	Intervention (*n* = 34)	
Sex				0.121
Male	27 (40.9%)	37.0	63.0	
Female	39 (59.1%)	56.4	43.6	
Age				0.729
Child	10 (15.2%)	40.0	60.0	
Adolescent	24 (36.4%)	54.2	45.8	
Adult	32 (48.5%)	46.9	53.1	
BMI at baseline				**0.026**
Underweight	5 (7.6%)	40.0	60.0	
Normal weight	24 (36.4%)	70.8	29.2	
Overweight	17 (25.8%)	23.5	76.5	
Obesity	20 (30.3%)	45.0	55.0	
Weight objective achieved				**0.001**
Yes	34 (48.5%)	25.0	75.0	
No	32 (51.5%)	70.6	29.4	
Migrant origin				0.072
No	12 (18.2%)	25.0	75.0	
Yes	54 (81.8%)	53.7	46.3	
Educational status				**0.015**
Low	55 (83.3%)	41.8	58.2	
Medium or high	11 (16.7%)	81.8	18.2	
Employment status				**0.011**
Unemployed	46 (69.7%)	37.8	62.2	
Employed	20 (30.3%)	71.4	28.6	
Family structure				**0.018**
Traditional	42 (63.6%)	29.2	70.8	
Non-traditional	24 (36.4%)	59.5	40.5	

Statistical significance was defined as *p* < 0.05. Significant results are shown in bold font.

**Table 2 nutrients-14-04980-t002:** Mean and SD values of main parameters before and after the intervention (children and adolescents).

*n* = 34	Pre-Intervention T0 (Mid-October 2021)			Post-Intervention T1 (End of December 2021)			Mean Differences	
	Control = 17	Intervention = 17		Control = 17	Intervention = 17		Control Intervention	
	Mean (SD)	Mean (SD)	*p*-Value	Mean (SD)	Mean (SD)	*p*-Value	MD (T1-T0)	MD (T1-T0)
Age (years)	11.64 (3.88)	11.54 (4.28)	0.934	11.90 (3.88)	11.80 (4.28)	0.934	-	-
ADM	9.82 (3.00)	9.76 (3.36)	0.957	11.58 (1.87)	10.94 (1.51)	0.277	1.76	1.18
Weight (kg)	44.20 (19.29)	48.30 (24.97)	0.596	45.48 (19.99)	47.97 (24.17)	0.327	1.28	−0.33
Height (m)	147.73 (0.22)	148.24 (0.22)	0.948	147.95 (0.23)	148.49 (0.23)	0.953	0.22	0.25
BMI (kg/m^2^)	19.28 (4.74)	20.65 (7.04)	0.511	19.72 (6.66)	20.50 (6.86)	0.704	0.44	−0.15
BMI z-score	0.00 (1.28)	0.14 (1.68)	0.781	0.12 (1.28)	0.13 (1.60)	0.975	0.12	−0.01
Fat (kg)	10.97 (6.52)	13.5 (10.30)	0.433	12.61 (8.50)	14.23 (12.15)	0.655	1.64	0.73
Fat Mass Index	7.20 (3.76)	8.75 (7.02)	0.427	8.23 (4.93)	9.20 (7.32)	0.653	1.03	0.45
Water (percentage)	55.28 (4.80)	54.28 (7.62)	0.650	53.88 (5.78)	53.51 (7.76)	0.876	−1.40	−0.77
Lean mass (kg)	30.42 (14.43)	32.91 (14.38)	0.618	31.15 (13.14)	31.96 (13.40)	0.861	6.15	5.59
Waist (cm)	64.92 (10.42)	67.80 (15.90)	0.536	65.54 (10.53)	67.32 (15.42)	0.697	0.62	−0.48
Waist/height	0.44 (0.05)	0.45 (0.07)	0.570	0.44 (0.05)	0.45 (0.07)	0.789	0	0
Hip (in cm)	81.54 (15.67)	83.54 (20.97)	0.755	82.57 (16.65)	83.68 (20.31)	0.863	1.03	0.14
Uric acid (mg/dL)	4.42 (0.99)	4.25 (1.10)	0.655	4.04 (1.08)	4.14 (1.15)	0.812	−0.38	−0.11
Urea (mg/dL)	26.46 (6.23)	26.86 (5.99)	0.856	27.40 (5.86)	25.93 (4.87)	0.455	0.94	−0.93
GOT (U/L at 37 °C)	33.13 (7.16)	35.68 (11.48)	0.467	33.53 (10.95)	35.68 (9.84)	0.569	0.40	0
GPT (U/L at 37 °C)	17.66 (6.28)	17.93 (5.57)	0.900	18.73 (8.63)	18.37 (6.11)	0.894	1.07	0.44
GGT (U/L at 37 °C)	16.73 (3.41)	17.75 (3.76)	0.438	16.06 (1.70)	16.75 (3.78)	0.520	−0.67	−1.00
TC (mg/dL)	172.33 (28.78)	150.81 (24.99)	**0.034**	172.28 (26.61)	149.59 (24.05)	**0.020**	−0.05	−1.22
TG (mg/dL)	62.53 (24.21)	74.12 (42.79)	0.365	67.64 (26.88)	70.93 (37.55)	0.787	5.11	−3.19
HDL (mg/dL)	54.40 (8.99)	48.62 (10.41)	0.110	58.28 (10.29)	52.06 (11.45)	0.131	3.88	3.44
LDL (mg/dL)	105.26 (26.28)	87.37 (21.83)	**0.048**	100.42 (18.78)	83.37 (21.71)	**0.030**	−4.84	−4
Glucose (mg/dL)	87.66 (8.38)	83.43 (8.32)	0.170	86.26 (7.33)	84.25 (7.43)	0.454	−1.40	0.82
HbA1c (%)	5.27 (0.68)	4.82 (0.27)	**0.021**	5.36 (0.22)	5.18 (0.29)	0.071	0.09	0.36
SBP (mmHg)	115.14 (9.96)	113.47 (12.78)	0.673	108.67 (10.55)	110.91 (12.92)	0.585	−6.47	−2.56
DBP (mmHg)	68.58 (3.67)	63.79 (9.33)	0.057	68.76 (9.06)	71.76 (9.59)	0.356	0.18	7.97
Heartrate (beats per minute)	79.55 (14.45)	81.85 (14.28)	0.850	80.17 (12.64)	76.91 (14.06)	0.502	−0.53	2.3

ADM: Adherence to the Mediterranean Diet; BMI: Body Mass Index; DBP: Diastolic Blood Pressure; HDL: high-density lipoprotein cholesterol; HbA1c: glycated hemoglobin A1c; GOT: glutamic oxaloacetic transaminase; GGT: Gamma-glutamyl transferase; GPT: Glutamate-Pyruvate Transaminase; MD: Mean difference; SBP: Systolic Blood Pressure; SD: Standard Deviation; TG: triglycerides, TC: total cholesterol; T0: baseline; T1: follow-up. Statistical significance was defined as *p* < 0.05. Significant results are shown in bold font.

**Table 3 nutrients-14-04980-t003:** Mean and SD values of main parameters before and after the intervention (adults).

*n* = 32	Pre-Intervention (Mid-October 2021)			Post-Intervention (End of December 2021)		
	Control = 15	Intervention = 17		Control = 15	Intervention = 17	
	Mean (SD)	Mean (SD)	*p*-Value	Mean (SD)	Mean (SD)	*p*-Value
Age (years)	42.60 (13.80)	44.29 (13.53)	0.350	42.84 (13.80)	44,53 (13.53)	0.708
Adherence to Mediterranean Diet	8.73 (2.37)	6.47 (2.42)	0.012	10.21 (2.11)	10.35 (1.22)	0.821
Weight (kg)	69.40 (14.75)	87.02 (18.96)	0.007	70.31 (15.32)	85.44 (18.66)	0.021
Height (m)	160.06 (0.08)	169.69 (9.43)	0.005	160.06 (0.08)	169.69 (9.43)	0.005
BMI (kg/m^2^)	27.20 (5.90)	30.28 (6.31)	0.166	27.56 (6.09)	29.69 (6.05)	0.704
Fat (kg)	22.92 (10.60)	30.35 (13.11)	0.070	23.93 (11.07)	29.65 (12.62)	0.195
Fat Mass Index	14.09 (6.54)	18.01 (7.92)	0.427	14.05 (7.81)	17.57 (7.59)	0.653
Water (percentage)	48.91 (7.40)	46.21 (7.35)	0.269	48.60 (7.09)	46.81 (7.33)	0.500
Lean mass (kg)	44.97 (7.52)	53.51 (9.72)	0.007	43.83 (7.57)	52.98 (9.85)	0.008
Waist (cm)	84.51 (13.98)	94.90 (15.90)	0.025	85.10 (14.95)	93.50 (11.95))	0.093
Waist/height	0.52 (0.09)	0.56 (0.07)	0.249	0.49 (0.16)	0.55 (0.06)	0.238
Hip (in cm)	100.94 (11.62)	111.47 (13.44)	0.025	94.23 (28.39)	109.45 (12.05)	0.053
Uric acid (mg/dL)	4.52 (0.00)	5.33 (1.08)	0.024	4.19 (1.08)	5.17 (1.15)	0.031
Urea (mg/dL)	30.20 (10.40)	36.78 (8.76)	0.066	33.78 (14.68)	33.43 (8.57)	0.936
GOT (U/L at 37 °C)	31.26 (12.14)	29.94 (11.36)	0.752	32.21 (11.17)	27.76 (5.79)	0.164
GPT (U/L at 37 °C)	25.73 (16.60)	26.52 (13.55)	0.882	22.78 (12.42)	21.70 (10.25)	0.793
GGT (U/L at 37 °C)	25.73 (15.91)	28.00 (10.54)	0.635	23.92 (16.47)	26.76 (12.50)	0.590
TC (mg/dL)	190.53 (32.46)	188.94 (35.96)	0.897	191.92 (48.77)	187.47 (28.51)	0.753
TG (mg/dL)	103.40 (61.43)	137.41 (74.41)	0.172	130.50 (82.64)	111.11 (53.89)	0.438
HDL (mg/dL)	53.60 (15.00)	48.17 (11.95)	0.264	55.50 (18.83)	47.78 (12.76)	0.185
LDL (mg/dL)	114.73 (30.34)	115.41 (27.27)	0.947	113.85 (42.22)	118.41 (21.94)	0.702
Glucose (mg/dL)	96.06 (16.33)	97.17 (15.77)	0.846	97.23 (20.22)	97.29 (15.88)	0.992
HbA1c (%)	5.65 (1.00)	5.27 (0.47)	0.200	5.95 (0.86)	5.58 (0.46)	0.171
SBP (mmHg)	128.96 (24.94)	138.41 (22.31)	0.381	129.75 (21.68)	138.70 (17.11)	0.215
DBP (mmHg)	69.73 (8.63)	78.73 (12.33)	0.025	80.00 (11.65)	86.58 (10.86)	0.115
Heartrate (beats per minute)	72.90 (9.56)	75.23 (8.14)	0.850	71.71 (9.55)	76.02 (9.46)	0.502

ADM: Adherence to the Mediterranean Diet; BMI: Body Mass Index; DBP: Diastolic Blood Pressure; HDL: high-density lipoprotein cholesterol; HbA1c: glycated hemoglobin A1c; GOT: glutamic oxaloacetic transaminase; GGT: Gamma-glutamyl transferase; GPT: Glutamate-Pyruvate Transaminase; MD: Mean difference; SBP: Systolic Blood Pressure; SD: Standard Deviation; TG: triglycerides, TC: total cholesterol.

**Table 4 nutrients-14-04980-t004:** Mean differences comparing a 10-week educational program for participants in the intervention and the control group divided by children and adolescents vs. adults.

N = 66	Children and Adolescents = 34				Adults = 32			
	MD (T1-T0) Control	MD (T1-T0) Intervention	MD Change (I-C)	*p*-Value	MD (T1-T0) Control	MD (T1-T0) Intervention	MD Change (I-C)	*p*-Value
ADM	1.76	1.18	−0.58	0.499	1.50	3.88	2.38	**0.016**
Weight (kg)	1.28	−0.33	−1.61	**0.002**	0.91	−1.58	−2.49	**0.017**
Height (m)	0.22	0.25	0.03	0.341	-	-	-	**-**
BMI (kg/m^2^)	0.44	−0.15	−0.59	0.499	0.36	−0.59	−0.95	**0.019**
BMI z-score	0.12	−0.01	−0.13	0.066	-	-	-	
Fat (kg)	1.63	0.67	−0.96	0.136	1.01	−0.70	−2.21	**0.012**
Fat Mass Index	1.03	0.45	−0.58	0.128	−0.04	−0.44	−0.4	0.601
Water (percentage)	−1.40	−0.77	0.63	0.268	−0.31	0.60	0.91	0.152
Lean mass (kg)	0.73	−0.95	−1.68	0.184	−1.14	−0.53	0.61	0.334
Waist (cm)	0.62	−0.48	−1.10	0.075	0.59	−1.40	−1.99	0.108
Waist/height	0.01	0.01	0.00	0.091	−0.03	−0.01	0.02	0.249
Hip (in cm)	1.03	0.14	−0.89	0.120	−6.71	−2.02	4.69	0.439
Uric acid (mg/dL)	−0.38	−0.11	0.27	0.135	−0.32	−0.16	0.12	0.507
Urea (mg/dL)	0.94	−0.93	−1.87	0.367	2.93	−3.35	−6.27	**0.030**
GOT (U/L at 37 °C)	0.40	0.00	−0.40	0.905	0.21	−2.18	−2.39	0.360
GPT (U/L at 37 °C)	1.07	0.44	−0.63	0.798	−3.57	−4.82	−1.87	0.668
GGT (U/L at 37 °C)	−0.67	−1.00	−0.33	0.805	−1.81	−1.24	0.55	0.847
Cholesterol (mg/dL)	−0.05	−1.31	−1.26	0.864	1.39	−1.47	−2.86	0.914
TG (mg/dL)	5.11	−3.19	−8.30	0.321	27.10	−26.30	−53.40	**0.023**
HDL (mg/dL)	1.88	3.44	−0.44	0.933	1.90	−0.41	−2.31	0.513
LDL (mg/dL)	−4.84	−4.00	0.84	0.576	−0.88	3.00	3.88	0.446
Glucose (mg/dL)	−1.4	0.82	2.22	0.460	1.17	0.12	−1.05	0.749
HbA1c (%)	0.09	0.36	0.27	0.236	0.30	0.31	0.01	0.595
SBP (mmHg)	−6.47	−2.56	3.91	0.224	1.14	0.29	−0.85	0.752
DBP (mmHg)	0.18	7.97	7.79	**0.014**	10.27	7.85	−2.42	0.417
Heartrate (beats per minute)	0.62	−4.94	−5.56	0.077	−1.19	0.79	1.98	0.717

ADM: Adherence to the Mediterranean Diet; BMI: Body Mass Index; DBP: Diastolic Blood Pressure; HDL: high-density lipoprotein cholesterol; HbA1c: glycated hemoglobin A1c; GOT: glutamic oxaloacetic transaminase; GGT: Gamma-glutamyl transferase; GPT: Glutamate-Pyruvate Transaminase; MD: Mean difference; SBP: Systolic Blood Pressure; SD: Standard Deviation; TG: triglycerides, TC: total cholesterol; T0: baseline; T1: follow-up. Statistical significance was defined as *p* < 0.05. Significant results are shown in bold font.

**Table 5 nutrients-14-04980-t005:** Distribution of the frequency of purchase of the different food groups by family unit (in percentage).

	Control Group			Intervention Group		
	First Month Intervention	Last Month Intervention	Post-Intervention	First Month Intervention	Last Month Intervention	Post-Intervention
Cereals & cereal products	2.90	1.48	1.06	2.99	2.67	0.50
Pasta & rice	2.50	2.80	3.87	4.04	5.34	2.85
Ready-to-eat dishes	6.45	8.07	5.11	2.81	2.81	1.12
Dairy & dairy-free products	9.02	5.77	7.75	7.02	5.49	6.07
Egg & egg dishes	2.66	3.62	2.82	2.72	4.50	2.85
Legumes	1.93	1.32	1.76	1.93	3.66	1.98
Meat & meat products	4.75	4.78	9.68	10.80	10.55	15.24
Fish & fish dishes	5.16	5.44	4.40	9.31	12.80	4.09
Fruits and vegetables	23.85	26.19	25.88	29.85	32.49	31.97
Salty snacks	2.10	3.13	1.76	0.35	0.00	1.86
Nuts & seeds	3.30	5.77	3.17	7.64	6.33	4.21
Sugar, bakery products, cakes, and confectionery	20.55	23.72	20.95	13.96	9.85	17.47
Commercial sauces	1.93	1.32	2.11	0.70	0.42	1.73
Sweetened beverages	8.94	5.77	8.80	2.90	1.41	7.06
Unsaturated oils (olive oil)	2.99	1.69	0.99	3.95	0.82	0.88

## Data Availability

Not applicable.

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
