# Peer review of "The Effect of Food Vouchers and an Educational Intervention on Promoting Healthy Eating in Vulnerable Families: A Pilot Study"

_nutrients, 2022, doi:10.3390/nu14234980_

Round 1
Reviewer 1 Report
The manuscript is well-written, however, the following points needs to be addressed in the revised version:
1. The 'Introduction' section is not strong enough to justify the study. Therefore, more pieces of evidence are necessary. Add a literature review part here, clearly indicate the research gaps, and justify your study on how this is going to solve an existing problem.
2. In the 'Materials and Methods', a figure visualizing the research framework would be helpful to understand the study design.
3. Though the findings are compared slightly with the existing studies, a more detailed discussion of the literature would strengthen the findings.
4. The future directions from this study should be included in the 'Conclusion' section.
5. Importantly, recommendations on the specific points of policy options, extension, or further research should be clearly presented.
Author Response
Reviewer #1:
The manuscript is well-written, however, the following points needs to be addressed in the revised version:
- The 'Introduction' section is not strong enough to justify the study. Therefore, more pieces of evidence are necessary. Add a literature review part here, clearly indicate the research gaps, and justify your study on how this is going to solve an existing problem.
Answer: Thank you very much. The following paragraphs (in red) have been added in the introduction section to justify our study:
“Food vouchers have been used as a tool to reduce health inequalities and improve barriers to a healthy diet. However, the results on the effects of these food subsidy programmes on vulnerable populations are controversial. Indeed, while some studies suggest that the mere provision of food vouchers is sufficient to improve dietary habits, others suggest that the possible positive effect is mediated by some educational intervention conducted simultaneously [16-18].
In addition, many of the previous studies have included food vouchers targeted only to the purchase of certain products (e.g. only fruit and vegetables) with less financial support. The lack of an educational intervention together with a financial aid could also be one of the possible reasons for the inconclusive results in favour of these social interventions [17, 19].
Moreover, most of the studies including an intervention in vulnerable groups have evaluated the effect that either education or incentives can have on their dietary patterns assessed subjectively with food frequency questionnaires (FFQ) or 24 hour-dietary recalls (24-HDR) instead of using more accurate tools [17]. Thus, these dietary assessment methods are generally limited by self-report biases [16]. The present paper use electronically recorded, supermarket-obtained transactions as an objective measure of family food purchases.
Therefore, the aim of this study was to assess if a 10-week intervention, including both food vouchers and nutritional and health education, improves diet quality and consequently the health of vulnerable participants in Zaragoza (Spain). Additionally, both groups (intervention and control) received, 8-weeks after the above 10-week intervention, only food vouchers (no educational intervention) to assess whether the possible positive effect of this intervention was maintained over time”.
- In the 'Materials and Methods', a figure visualizing the research framework would be helpful to understand the study design.
Answer: Thank you for your suggestion. We have included the following image:
Figure 1: Study design
- Though the findings are compared slightly with the existing studies, a more detailed discussion of the literature would strengthen the findings.
Answer: We have discussed more in detail our results and the findings from previous literature. As a result, these two new paragraphs were included:
“Our results showed that the food vouchers alone were not sufficient to improve the diet or health parameters analyzed for the extremely vulnerable groups. These results are in agreement with a previous study carried out in a low-income population of France. This study conducted a 3-month program assessing the effect of food voucher (only exchangeable for fresh fruit and vegetables) in the intake of fruits and vegetables and plasma levels of biomarkers. The group receiving both the food vouchers and the nutritional advise did not show either a higher increase in mean intake or a change in relevant biomarker levels compared with the group-receiving only dietary advice. The results of this French study suggest the importance of dietary advice over food vouchers [35].
Nonetheless, it should be borne in mind that other studies have pointed out that providing nutrition education only or food vouchers only has limited effects on health or the feeding practices of the participants. Indeed, in a recent study conducted in Ethiopia when provided both education and food vouchers together, child-feeding practices improved, and stunting prevalence decreased [36]”.
- The future directions from this study should be included in the 'Conclusion' section.
Answer: The following paragraph has been included in the conclusion section:
“Future lines of research should include longer-term educational intervention and follow-up to obtain more accurate results for health policy makers with objective dietary assessment tools. It is also expected that these studies will include greater tailoring and adaptation to the needs of these vulnerable groups”.
- Importantly, recommendations on the specific points of policy options, extension, or further research should be clearly presented.
Answer: Thank you for your suggestion. We have included the following paragraph in the conclusion section:
“In particular, our results suggest that public actions aimed at improving dietary patterns, achieving greater adherence to the Mediterranean diet or improving the health of participants should include active educational interventions (not merely passive ones such as handing out information leaflets), probably for six months or more, including follow-up and with the provision of food vouchers preferably restricted to the purchase of certain products (e.g. fruit, vegetables, pulses, nuts, dairy products, olive oil...)”.

Reviewer 2 Report
The entire family unit was randomized, resulting in 32 individuals in the 97 control group and 34 individuals in the intervention group.
You randomized within the household unit?! So one member could be in the treatment group, and another member could be in the control group? If so, how can you possibly assume a member in the control group did not actually also receive treatment?
The total sum of the score assigned to 152 each answer lead to the following result: low adherence = score from 0 to 5, average ad-153 herence = score from 6 to 9 and maximum adherence = score ≥10.
Why do you create three categories? You lose so much information and variability in the data. You go from a continuous variable with 0-14 points to three binary variables.
After the educational pro-296 gram, the mean weight of the subjects in the intervention group decreased, while those in 297 the control group gained weight.
In the table you show how the difference is statistically significant, but it already was before the study. What about the difference in weight within each group? So is the difference in the weight in treated individuals statistically significant or not?
The control group received food vouchers for 14 weeks (from mid-104 October to December 2021 and additionally for one more month in March 2022).
Were participants able to use the vouchers on any food/drink products? Or only “healthy” food/drink products?
Author Response
The entire family unit was randomized, resulting in 32 individuals in the control group and 34 individuals in the intervention group.
You randomized within the household unit?! So one member could be in the treatment group, and another member could be in the control group? If so, how can you possibly assume a member in the control group did not actually also receive treatment?
Answer: Thank you very much for pointing this out. Only one person of the household unit (usually the mother or the woman if they were a couple) was randomized and included in either the control group or the intervention group. Then all persons belonging to that household were automatically included in the control group or the intervention group. So, the whole household was always in the same group (control or intervention). To make it clearer we have included the following paragraph in the methods sections:
“Participants were randomly selected using the free OxMaR software for randomization of studies. Only one person of the household unit (usually the mother or the woman if they were a couple) was randomized and included in either the control group or the intervention group. Then all members belonging to that household were automatically included in the control group or the intervention group. Thus, the whole household unit was always in the same group (either control or intervention). Finally, 32 individuals were assigned to the control group and 34 individuals to the intervention group”.
The total sum of the score assigned to each answer lead to the following result: low adherence = score from 0 to 5, average adherence = score from 6 to 9 and maximum adherence = score ≥10.
Why do you create three categories? You lose so much information and variability in the data. You go from a continuous variable with 0-14 points to three binary variables.
Answer: Thank you for your remark. For all the analyses we used the continuous variables of adherence to the Mediterranean diet but we wanted to give the meaning of the categories for the readers low adherence = score from 0 to 5, average adherence = score from 6 to 9 and maximum adherence = score ≥10. We deleted this sentence, and we included the following paragraph (in red):
“To determine the degree of adherence to the Mediterranean diet in adults, a specific short questionnaire validated for the Spanish population and used by the Prevention with Mediterranean Diet (PREDIMED) group was used [20]. A trained dietitian administered this 14-item questionnaire. Briefly, for each item a score 1 and 0 was assigned. Once the values obtained for the 14 items were added up. Thus, the score ranged from 0 to 14 in which 0 points meant null adherence and the 14 points meant the highest adherence in adults”.
After the educational program, the mean weight of the subjects in the intervention group decreased, while those in the control group gained weight. In the table you show how the difference is statistically significant, but it already was before the study. What about the difference in weight within each group? So is the difference in the weight in treated individuals statistically significant or not?
Answer: Thank you for your comment. In children and adolescents, there was no statistically significant differences between the control and the intervention group at baseline. In adults, there was a statistically significant difference in weight and height but not in body mass index (BMI) at baseline. It should be noted that at baseline participants in the intervention group were almost 10 cm taller than those in the control group, so these significant differences in weight were largely due to the difference in height between the two groups. After the intervention, the change in the mean difference was significant for both children and adolescents, and adults in weight and BMI only in adults. So, this difference in weight in treated individuals was statistically significant.
The control group received food vouchers for 14 weeks (from mid-October to December 2021 and additionally for one more month in March 2022). Were participants able to use the vouchers on any food/drink products? Or only “healthy” food/drink products?
Answer: Thank for your comment. In the present study, no restriction regarding foods and drinks were conducted. However, participants were recommended to buy healthy food, and to avoid alcohol and sugar sweetened beverages (SSB).
This information was already in the manuscript (methods section and discussion section) and to make it clearer we added the following information in red.
“For that reason, all participants (from both, the intervention, and the control group) received unrestrictive voucher cards to obtain food or drinks at any supermarket of the DIA group in Zaragoza to finance family meals for 14 weeks (from mid-October to December 2021 and March 2022).
“In our study, no restriction regarding the food and drinks that could be chosen by participants were done. However, participants were recommended to buy healthy food, and to avoid alcohol and sugar sweetened beverages (SSB)”.

Round 2
Reviewer 1 Report
I am satisfied with the revision made by the authors. The revised manuscript can be accepted for publication.
Reviewer 2 Report
Good job.